# Mapping transgene insertion sites reveals the $\alpha$-Cre transgene expression in both developing retina and olfactory neurons

Yimeng Fan[1,2,4], Wenyue Chen[1,2,4], Ran Wei[1,2,4], Wei Qiang[1,2], Joel D. Pearson[3], Tao Yu[3], Rod Bremner [3✉] & Danian Chen [1,2,3✉]

The $Tg(Pax6\text{-}cre,GFP)^{2Pgr}$ ($\alpha$-Cre) mouse is a commonly used Cre line thought to be retinal-specific. Using targeted locus amplification (TLA), we mapped the insertion site of the transgene, and defined primers useful to deduce zygosity. Further analyses revealed four tandem copies of the transgene. The insertion site mapped to clusters of vomeronasal and olfactory receptor genes. Using $R26R$ and $Ai14$ Cre reporter mice, we confirmed retinal Cre activity, but also detected expression in $G\alpha_O^+$ olfactory neurons. Most $\alpha$-Cre$^+$ olfactory neurons do not express Pax6, implicating the influence of neighboring regulatory elements. RT-PCR and buried food pellet test did not detect any effects of the transgene on flanking genes in the nasal mucosa and retina. Together, these data precisely map $\alpha$-Cre, show that it does not affect surrounding loci, but reveal previously unanticipated transgene expression in olfactory neurons. The $\alpha$-Cre mouse can be a valuable tool in both retinal and olfactory research.

[1] Research Laboratory of Ophthalmology and Vision Sciences, State Key Laboratory of Biotherapy, West China Hospital, Sichuan University, Chengdu, China. [2] Department of Ophthalmology, West China Hospital, Sichuan University, Chengdu, China. [3] Lunenfeld-Tanenbaum Research Institute, Sinai Health System, and Departments of Ophthalmology and Visual Science, and Laboratory Medicine and Pathobiology, University of Toronto, Toronto, ON, Canada. [4]These authors contributed equally: Yimeng Fan, Wenyue Chen, Ran Wei. ✉email: bremner@lunenfeld.ca; danianchen2006@qq.com

Cre/LoxP technology is widely used in the field of mouse genetics. Both spatial and temporal genetic manipulations can be performed using cell type-specific enhancers/promoters to drive Cre recombinase expression in combination with Cre recognition (loxP) sites in interest genes[1]. Many retinal cell-targeted Cre transgenic mice have been created to facilitate the study of retinal development and retinal diseases[2]. The *Tg(Pax6-cre,GFP)2Pgr* (*α-Cre*) mouse (MGI:3052661) is a commonly used retina-specific Cre line[3]. In addition to the *α-enhancer*, the *α-Cre* transgene construct also has the *P0 promoter* of the *Pax6* gene, an *IRES-GFP* cassette, an intron from the *Hbb* (β-hemoglobin) gene, and an *SV40 poly (A)* sequence[3]. The *α-Cre* transgene is active from embryonic day 10 in peripheral retinal progenitors that give rise to all cells in the mature retina of this region[3,4]. In the mature eye, the *α-Cre* is also active in cells of the ciliary body[3,4] and GABAergic amacrine cells[5].

The *α-Cre* mouse line is generated via pronuclear microinjection of the *α-Cre* transgene construct[3]. Therefore, the integration site of the *α-Cre* transgene is random and unknown. Such insight is helpful to deduce whether the transgene might have deleterious effects, and to define primer pairs around the integration site to determine whether breeders are heterozygous or homozygous for the transgene. Previously, we reported that *Rb* gene knockout (*KO*) in the retina with *α-Cre* (*α-Cre; Rb^{f/f}*) induces ectopic division and apoptosis of many retinal excitatory neurons[6]. In order to elucidate the mechanism of *RbKO*-induced retinal cell death, we bred *α-Cre;Rb^{f/f}* mice with *Bax^{−/−}* mice[7]. Bax is a member of Bcl2 family proteins and mediates neuronal cell death[8], including physiological retinal apoptosis[9,10] and neuronal death in the *Rb/p107* double knockout brain[11]. *Bax*-null males are infertile[12]. To generate *α-Cre; Rb^{f/f};Bax^{−/−}* mice, we first generated *α-Cre;Rb^{f/f};Bax^{+/−}* males and *Rb^{f/f};Bax^{−/−}* females, and inter-bred them. Of 96 pups, we obtained 50 pups with *α-Cre* and 48 pups with *Bax^{−/−}* alleles as predicted, but only five pups (5.2%) with *α-Cre;Bax^{−/−}*, far less than the expected 24 pups (25%), suggesting a genetic linkage between the *α-Cre* transgene and the *Bax* locus on chromosome seven[7]. This result prompted us to investigate the precise location of the *α-Cre* transgenic

insertion site and any possible functional consequences of the genetic recombination.

Commonly used methods for identifying integration sites include inverse PCR[13–15], whole-genome sequencing and capture-based targeted re-sequencing[16,17]. However, these methods are inconvenient[18]. Recently, several studies have employed targeted locus amplification (TLA) to map integration sites of different *Cre* or other transgenes[18–22]. TLA selectively amplifies and sequences entire genes based on the crosslinking of physically proximal sequences and works without detailed prior locus information[23]. TLA can generate complex DNA libraries covering >100 kb of contiguous sequence surrounding one primer pair complementary to a short locus-specific sequence[24]. Inspired by these studies, especially the work identifying the insertion sites for three transgenes, which label some retinal cells[18] and for seven *Cre* transgenes[19], we decided to use TLA to search the insertion site of the *α-Cre* transgene.

In this study, we identified the genomic insertion site of the *α-Cre* transgene on mouse chromosome 7 by TLA analysis. We found it is among clusters of vomeronasal receptor 2 (*Vmn2r*) genes and olfactory receptor (*Olfr*) genes. Indeed, the *α-Cre* transgene is also expressed in many olfactory neurons. Notably, the *α-Cre* transgene has no significant effects on the expression of flanking genes and olfactory function.

## Results

**Mapping the integration site of the *α-Cre* transgene by TLA.** To identify the precise *α-Cre* integration site, we performed TLA analysis on spleen cells from P70 *α-Cre* mice. Two primer sets complementary to the Cre gene sequences were designed (Fig. 1a). Primer set 1 included primer RV461 and FW907; primer set 2 included primers RV330 and FW408 (Supplementary Table 1). Both were used to generate products that were sequenced to a depth of 2 Mb (Fig. 1b). We mapped the sequence data to the mouse genome (Version mm10) to analyze the integration site and to confirm the component sequences of the *α-Cre* transgene. TLA revealed that the *α-Cre* transgene is inserted

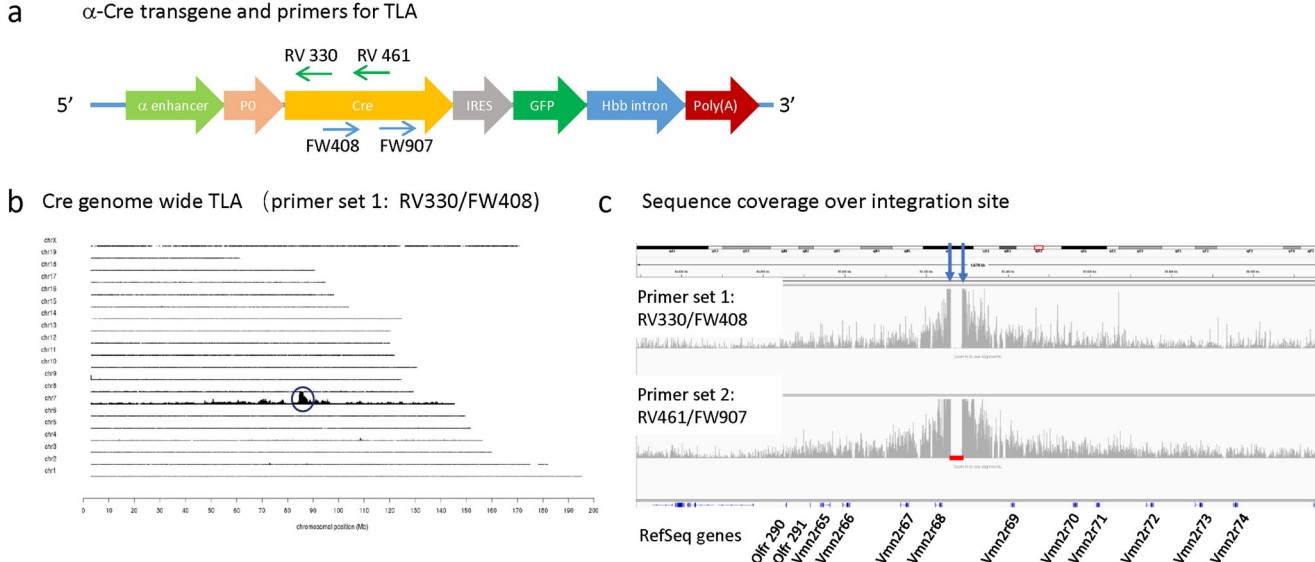

**Fig. 1 α-Cre locus identification. a** Schematic of the vector used to make α-Cre mice (including Pax6 α-enhancer, Pax6 P0 promoter, Cre, IRES, GFP, *Hbb* intron, and SV40 poly A sequence), showing positions of primers within Cre used for TLA. **b** TLA sequence coverage across the mouse genome using primer set 1. The chromosomes are indicated on the *y*-axis, the chromosomal position on the *x*-axis. Similar results were obtained with primer set 2. Peak at Chromosome 7 shows inserted sequence (black circle). **c** Regional coverage of the α-Cre insertion site on Chromosome 7 using both sets of primers, showing a 27 kb deletion and spanning 2 Mb. Some RefSeq genes flanking the insertion site (version Mm10) are also indicated.

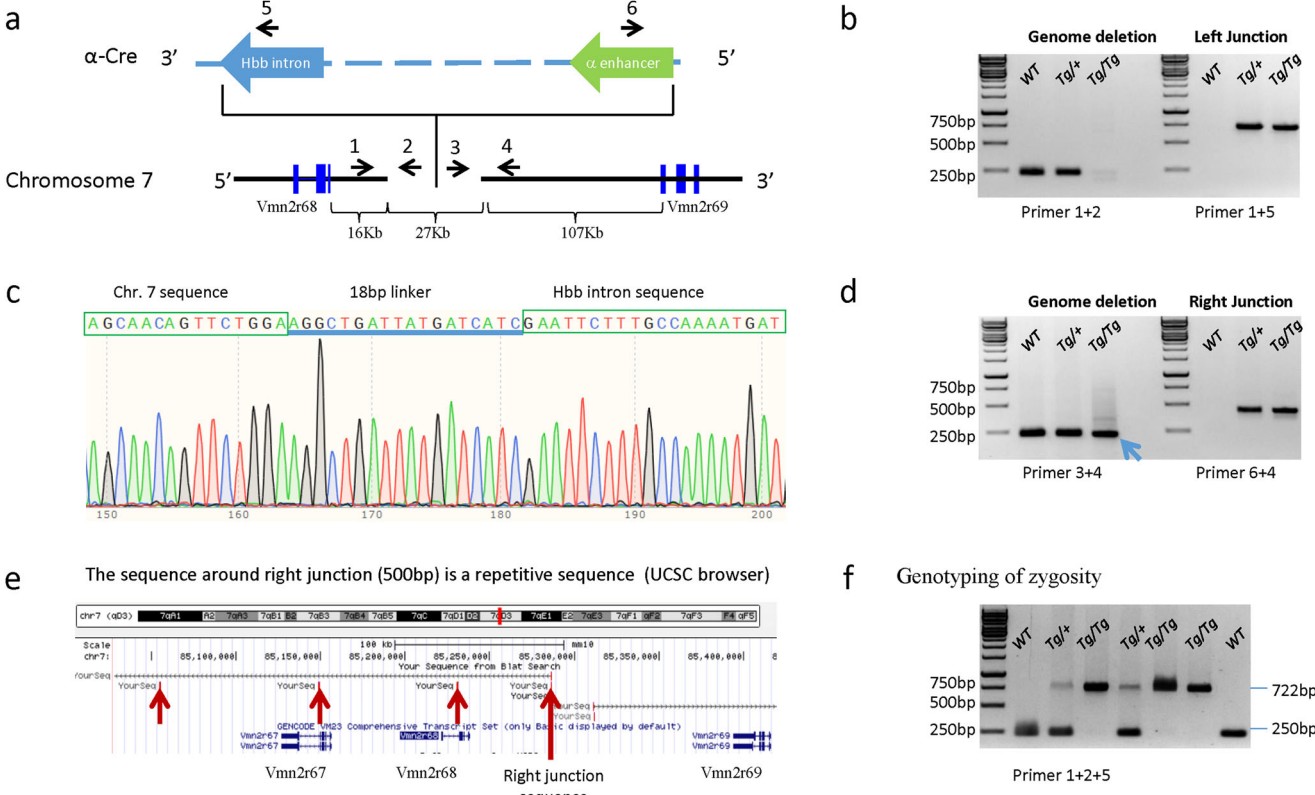

**Fig. 2 Confirmation of the transgene insertion site on Chromosome 7, and genotyping of zygosity. a** Map of the transgene/chromosome 7 left and right junctions. Primers were designed to confirm the left and right junctions between the transgene and Chromosome 7, as well as the predicted deletion. **b** Primers 1 and 2 were used to confirm the 27 kb deletion. The 250 bp band is absent in putative homozygous α-Cre animals. Primers 1 and 5 were used to confirm the left junction between the 3′ end of the α-Cre transgene and the 5′ end of Chromosome 7 in homozygous animals. This 722 bp band is absent in wild-type animals. **c** The chromatogram of the 722 bp band in **b** showing the chromosome 7 sequence linked to the *Hbb* intron through an 18 bp linker. No intervening SV40 poly A sequence was detected. **d** Primers 3 and 4 were designed to confirm the right-hand junction. However, the 250 bp band is still present in putative homozygous α-Cre animals (arrow). Primers 4 and 6 were used to confirm the right junction between the 5′ end of the α-Cre transgene and the 3′ end of Chromosome 7 in homozygous animals. This 436 bp band is absent in wild-type animals. **e** The 500 bp chromosome 7 sequence at the right junction of the α-Cre transgene is repetitive (version Mm10). Four such repeats are indicated around *Vmn2r67* and *Vmn2r68*, (red arrows). **f** Genotyping zygosity of the α-Cre transgene, using primers 1 + 2 + 5 from the left junction. Totally 7 animals were genotyped, including 2 wild types, 2 hemizygotes, and 3 homozygotes.

inversely into mouse chromosome 7 (Chr7: 85259496–85287099) and that the insertion is accompanied by a 27 kb deletion (Fig. 1c). There are no annotated genes in the integration region. No single nucleotide variant was seen in the Cre sequence. From the current dataset TG (transgene)-TG fusions could not be accessed.

To confirm the insertion, we designed primers flanking the predicted left and right junctions (Supplementary Table 2). We used genomic DNA from wild type, hemizygotes (*Tg*/+) and homozygotes (*Tg*/*Tg*) of the α-Cre transgene, determined by progeny testing assays, in these confirmations. Primer 1 targets the left junction but combining it with several primers from the *SV40 polyA* region (Supplementary Table 3) did not generate any specific PCR products (Fig. 2a and Supplementary Fig. 1). Primer 1 and primer 5 from the *Hbb* gene intron generated a 722 bp PCR product, which was absent in wild type animals (Fig. 2b). Sequencing of this 722 bp PCR product revealed that chromosome 7 sequence is linked through an 18 bp linker (AGGCT-GATTATGATCATC) to the *Hbb* intron (Fig. 2c), and not, as anticipated, through the *SV40 poly A* sequence (Fig. 1a). Primer 1 and 2 (from the deleted region) generated a 250 bp PCR product in wild type animals, but not α-Cre homozygotes (Fig. 2b).

Together these data reveal the precise left junction of the α-Cre transgene in chromosome 7, and provide a primer set that is useful to define zygosity.

To map the right junction, we used primer 4 from chromosome 7, and primer 6 from the α-enhancer (Fig. 2a), which generated a 436 bp PCR product in heterozygotes and homozygotes of the α-Cre transgene, but no product from wild type DNA (Fig. 2d). However, using primer 4 with primer 3 from the deleted region (Fig. 2a) generated a 250 bp PCR product both in wild type, heterozygotes and homozygotes of the α-Cre transgenes (Fig. 2d). This was unexpected, but Blast analysis using the UCSC genomic browser found that the 500 bp sequence around the right junction has multiple copies along chromosome 7 (Fig. 2e). These repetitive sequences are in introns or non-coding regions around *Vmn2r27-68* genes. These data confirm the chromosome 7 location of the right junction, but while primers 4 + 6 are useful to confirm transgene presence, the repetitive nature of the flanking sequence precludes the use of primers in this area (e.g., 3 + 4) to define zygosity. The left junction solves this issue: Primers 1 + 5 can confirm transgene presence, and Primers 1 + 2 can determine zygosity, as this amplicon is present in heterozygotes, but absent in homozygotes (Fig. 2a, b).

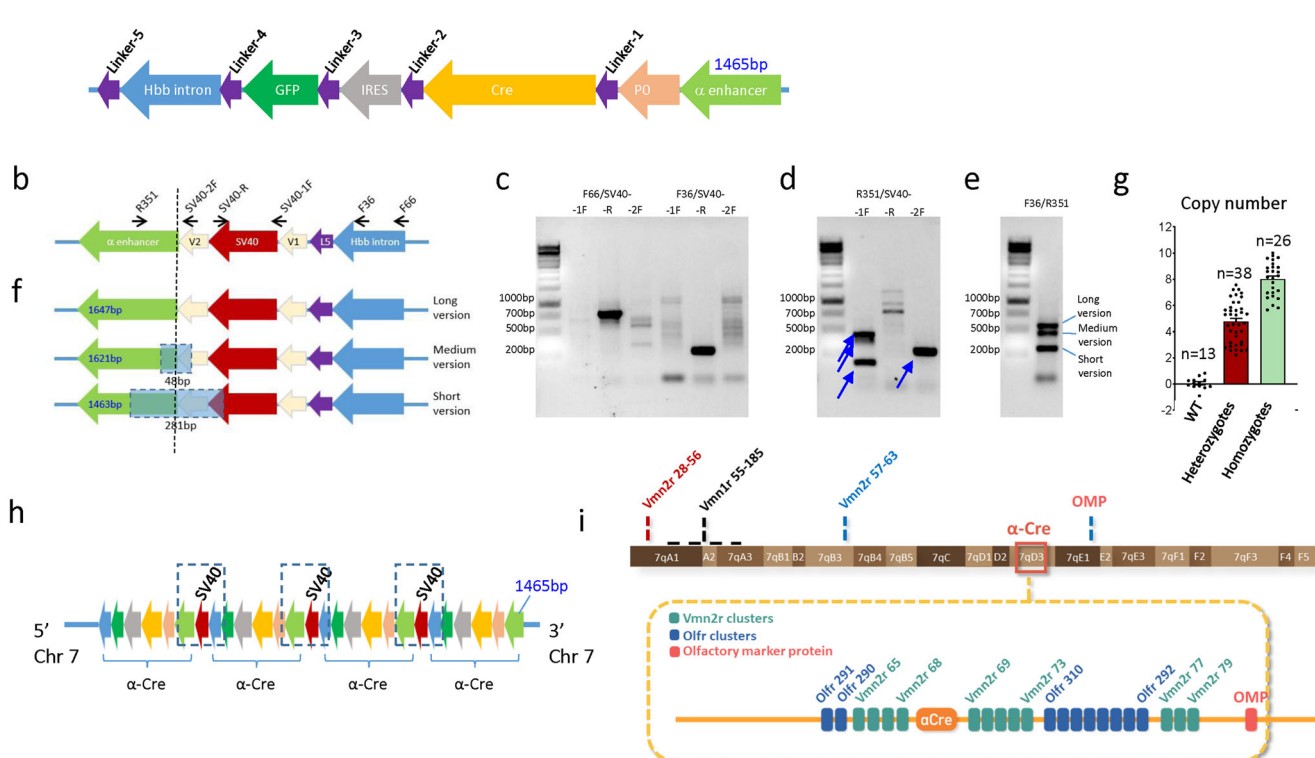

**Fig. 3 Evidence for four tandem transgene copies. a** Sequences from various PCR fragments were stitched together to deduce the arrangement of the α-Cre transgene, including the linker segments 1–5, but missing the SV40 polyA tail. The 1465 bp α enhancer shown here is adjacent to the 3′ side of chromosome 7. The position of the SV40 polyA tails and other tandem copies of the transgene were deduced in **b–h**. **b** Schematic showing the predicted junction between two tandem transgenes; the vertical dashed line indicates the junction site. The indicted primers were used to deduce the relationship between SV40 polyA signal and the *Hbb* intron or α-enhancer. **c** Primers from the *Hbb* intron (F66, F36) and SV40 polyA signal (SV40-1F, -2F, -R) confirmed that the SV40 signal is downstream of the *Hbb* intron. **d** Primers from the α-enhancer (R351) and SV40 polyA signal (SV40-1F, -2F, -R) confirmed the existence of tandem transgene copies. Blue arrows indicate multiple bands with R351/SV40-1F, but only one band with R351/SV40-2F primers. **e** Primers from *Hbb* intron (F36) and the α-enhancer (R351) confirmed three versions of the SV40/α-enhancer junction. **f** Schematic, based on sequencing of the PCR produces in **e** showing the relationship between long, medium and short versions of the SV40/α-enhancer junction. Dashed boxes (48 bp or 281 bp) indicate the deleted regions. **g** Quantitative PCR was used to measure transgene copy numbers in WT mice versus those that were heterozygous or homozygous for α-Cre. The data is mean ± SE. **h** Schematic showing 4 tandem copies of the α-Cre transgene in the chromosome 7 insertion site. The dashed boxes highlight the regions shown in more detail in **b**, **f**. Note that only three copies include the SV40 polyA signal, and the α-enhancer is a slightly different size in each copy (see sizes in **f**). Apart from the right-most copy, the order of these variants along the chromosome is undetermined. **i** Schematic showing α-Cre location relative to the entire chromosome and select chromosome 7 loci (top), and the neighboring *Vmnr2* and *Olfr* clusters within 7qD3 (bottom, dashed orange box). *Hbb*, hemoglobin beta gene; L5, linker 5; *OMP*, Olfactory marker protein gene; V1, V2 vector sequence 1 and 2 flanking the SV40 polyA signal; *Vmn1r*, vomeronasal receptor 1; *Vmn2r*, vomeronasal receptor 2; *Olfr*, olfactory receptor.

**Genotyping of zygosity of the α-Cre transgene**. Progeny testing is the gold standard method to deduce the zygosity of a transgene[25]. However this method is relatively expensive and time-consuming. With the precise insertion site and the sequences of the α-Cre transgene in hand, simple PCR reactions can be used to define zygosity. As mentioned above, primers from the left junction are suitable for this purpose (Supplementary Table 2, Fig. 2a). Using primers 1, 2, and 5 from the left junction of the insertion site, we could determine the zygosity easily (Fig. 2f). The 250 bp band is wild type, and the 722 bp band is the α-Cre transgene. Homozygotes only have the 722 bp, wild types only have the 250 bp band, and hemizygotes have both bands. Our results matched the progeny testing assays.

**Determining the SV40 polyA site and copy number of the α-Cre transgene**. Using multiple primer sets to amplify overlapping fragments (Supplementary Table 3), we sequenced the α-Cre transgene from the α enhancer to the *Hbb* intron, without the

SV40 polyA signal. The entire sequence of this stretch was 4911 bp, including 5 linkers (Fig. 3a). The lack of the SV40 polyA signal at the left junction with chromosome 7 suggests it was deleted during integration (Fig. 2c). TLA detected SV40 polyA sequence, but both flanking regions were vector sequences without any chromosome 7 elements, suggesting tandem integration.

To confirm the presence of SV40 polyA sequence and to test for tandem arrays, we designed primers in the *Hbb* intron, SV40 polyA tail and the α-enhancer (Fig. 3b). Combining forward primers in the *Hbb* intron (F66 and F36) with a reverse primer in the SV40 poly A tail (SV40-R) robustly amplified the expected fragments (Fig. 3c). In contrast, combining forward *Hbb* primers with forward SV40 primers (SV40-1F, SV40-2F) gave only weak non-specific bands (Fig. 3c). Similarly, a reverse primer in the α-enhancer generated specific PCR products with SV40-1F/2F primers, and not with SV40-R primer (Fig. 3d), supporting the existence of tandem transgene copies. Interestingly, several PCR products were obtained with R351/SV40-1F, but only one with

R351/SV40-2F (Fig. 3d), suggesting there were several forms of the *SV40-α enhancer* junction. This was confirmed with primers F36/R351 designed to amplify the entire *α-enhancer-SV40-Hbb* region, and which detected long, medium and short PCR products (Fig. 3e).

DNA sequencing these three amplicons revealed that, relative to the long PCR product, the medium version lacked 48 bp, which included part of vector sequence 2 (V2) adjacent to the *SV40 polyA* signal and part of the *α enhancer*, while the short version lacked 281 bp, which included part of the *SV40 polyA* signal, all of *V2*, and part of the *α-enhancer* (Fig. 3f). These differences at the junctions of tandem copies created variable *α-enhancer* sizes of 1647bp, 1621bp, and 1463 bp (Fig. 3f). The *α enhancer* sequence linked to chromosome 7 (Figs. 2a and 3a) is like the short version and is 1465 bp. These data indicate at least 4 copies of the *α-Cre* transgene, but only 3 copies of the *SV40 polyA* signal. Indeed, quantitative real-time PCR analysis of a 143 bp fragment from the Pax6 P0 promoter showed that the average transgene copy number is about 4.5 ($4.52 \pm 0.26$) for heterozygotes ($n = 38$), and 8.2 ($8.205 \pm 0.26$) for homozygotes ($n = 26$), respectively, (mean $\pm$ SE) (Fig. 3g). Multiple copies of transgenes are frequently integrated in a head-to-tail tandem array at a single genomic site[26]. In summary, our data indicate that there are four tandem copies of the *α-Cre* transgene, although further work would be required to deduce the exact order of the three versions of the *SV40-α enhancer* junction (Fig. 3h).

**Expression of the *α-Cre* transgene in the retina**. The *α-Cre* transgene integration site (Fig. 2) is among *Vmn2r* and olfactory receptor (*Olfr*) gene clusters (Fig. 3i). 4C (circular chromosome conformation capture) technology was first developed to study gene interactions between the *Hbb* gene on qE3 and other genes on chromosome seven[27]. They found that multiple *Olfr* genes on chromosome 7 interact with the *Hbb* gene, including those flanking the *α-Cre* transgene (Fig. 3i). Therefore, it is possible that regulatory elements of these olfactory receptor genes may induce *α-Cre* transgene expression in olfactory neurons. To address this issue, we crossed the *α-Cre* line with two commonly used Cre reporter lines, B6.Cg-Gt(ROSA)26Sor*tm14(CAG-tdTomato)Hze/J* mouse[28] (also referred to as *Ai14*), and Gt(ROSA)26Sor*tm1Sor*, also referred to as *R26R*[29] to obtain *α-Cre;Ai14* and *α-Cre;R26R* double transgenic mice. We then examined *α-Cre* expression pattern at different time points, first in the retina to confirm the expected patterns with these reporters, and then the olfactory system to search for ectopic transgene activation.

At embryonic day 12 (E12), horizontal sections through *α-Cre;Ai14* mouse heads confirmed, as expected, that α-Cre is active in nasal and temporal peripheral retina (Fig. 4a). Retinal ganglion cells and their axons (optic nerve and optic chiasm) were also tdTomato-positive. There was no expression in lens, cornea or any other tissue in the eye. X-gal staining of horizontal sections of *α-Cre;R26R* eyeballs revealed a similar pattern at E14 (Fig. 4b). At the day of birth (post-natal day 0, P0), nasal and temporal peripheral retina of *α-Cre;R26R* mice exhibited deep blue staining, while the dorsal and ventral retina had no staining (Fig. 4c). This was consistent with results from whole-mount analyses of *α-Cre;Ai14* retina at P18 (Fig. 4d).

Using anti-Cre, anti-GFP, and anti-mCherry antibody, we next compared the expression of Cre, GFP and tdTomato in *α-Cre;Ai14* retinas at E12. We found the area of Cre expression was similar to that of GFP expression, and the GFP/Cre expression areas were significantly smaller than that of tdTomato (Fig. 4e). GFP/Cre represents current expression level of the *α-Cre* transgene, while tdTomato marks lineage. Down-regulation of retinal *α-Cre* expression is well known[3], and we also confirmed that silencing begins even by E12, especially in retinal progenitor

cells close to the central retina. Cre and GFP exhibited a distal[high]-proximal[low] expression gradient (Fig. 4e), in accordance with the *α-enhancer*-dependent *Pax6* expression gradient across the optic cup[4].

At E16, the area of Cre expression in retinal progenitor cells was limited to the very peripheral retina, close to the ciliary body, and was smaller than that of GFP expression, which may reflect lower sensitivity of Cre detection, and/or a longer half-life of GFP protein versus Cre protein after the *Cre-IRES-GFP* mRNA is silenced (Fig. 4f). At E16, bright Cre and GFP was also evident at the inner side of the NBL (Fig. 4f). Co-staining with the amacrine cell marker AP2α showed that GFP was expressed in amacrine cells from E16 (Fig. 4g). These cells were Ki67 negative, suggesting they had exited the cell cycle (Fig. 4h). Spatially, GFP[+] amacrine cells were found across the whole retina without any apparent gradient suggesting it was controlled by the *Pax6 P0 promoter*, and as reported previously[3,4]. These data show that the *α-Cre* mice in our group behave as expected with respect to retinal expression.

**Expression of the *α-Cre* transgene in the olfactory system**. The murine olfactory system includes two distinct chemosensory systems, the main and accessory olfactory systems. The former has main olfactory sensory epithelium (MOE) and the septal organ of Masera, all projecting to the olfactory bulb (OB). The latter consists of the vomeronasal organ (VNO) and the ganglion of Grüneberg, all projecting to accessory olfactory bulb (AOB)[30,31]. Olfactory sensory neurons (OSNs) in these areas express different repertoires of chemosensory receptors, which can recognize thousands of odorant molecules or pheromones[30,31]. The major chemosensory receptors include olfactory receptors (Olfr) mainly expressed in MOE, and vomeronasal receptors (Vmnr) mainly expressed in VNO[31]. olfactory receptors belong to the family of rhodopsin-like G protein–coupled receptors (GPCRs)[32]. Mice have about 1,000 *Olfr* genes locating on almost every chromosome in clusters[32]. VNO sensory neurons express two families of GPCRs, Vmnr type 1 and 2 (Vmn1Rs and Vmn2Rs). Murine Vmn1Rs include about 180 GPCRs and Vmn2Rs include about 120 GPCRs. They are distantly related to bitter taste receptors and sweet taste receptors respectively. Vmn1Rs are expressed in apical VNO sensory neurons[32], while Vmn2Rs are expressed in basal VNO sensory neurons[33].

In coronal sections of the heads of P2 *α-Cre;Ai14* mice, there were many tdTomato[+] cells in both VNO and some areas of MOE (Fig. 5a). The Cre[+] cells always co-localized with the GFP[+] cells (Fig. 5b, white arrows), suggesting the IRES used in the original *α-Cre* transgene bicistronic construct (Fig. 1a) induced 1:1 expression between the first (*Cre*) and the second (*GFP*) genes in the MOE and VNO, even though there is report that the expression of the second gene is generally lower than the first gene[34]. However, like the retina (Fig. 4e), there were much less Cre[+] or GFP[+] cells than tdTomato[+] cells in the MOE and VNO, suggesting that the *α-Cre* transgene is also silenced in many MOE and VNO neurons (Fig. 5b). As a result, we only use tdTomato as the indicator of Cre activity in the following cell type and Pax6 expression studies.

The mouse VNO mainly has two types (apical and basal) vomeronasal sensory neurons (VSNs). Apical VSNs express the G protein Gαi2 and Vmn1Rs, basal VSNs express Gαo and Vmn2Rs[35]. Most VNO tdTomato+ cells in *α-Cre* mice were in the basal layer (Fig. 5a–d) and were co-localized with Gα₀ (Fig. 5c) but not Gαi2 (Fig. 5d). Thus, α-Cre is mainly expressed in basal VSNs, consistent with its insertion site in the *Vmn2r65-73* gene cluster (Fig. 3i). Like the optic nerve (Fig. 4a), VSN axons

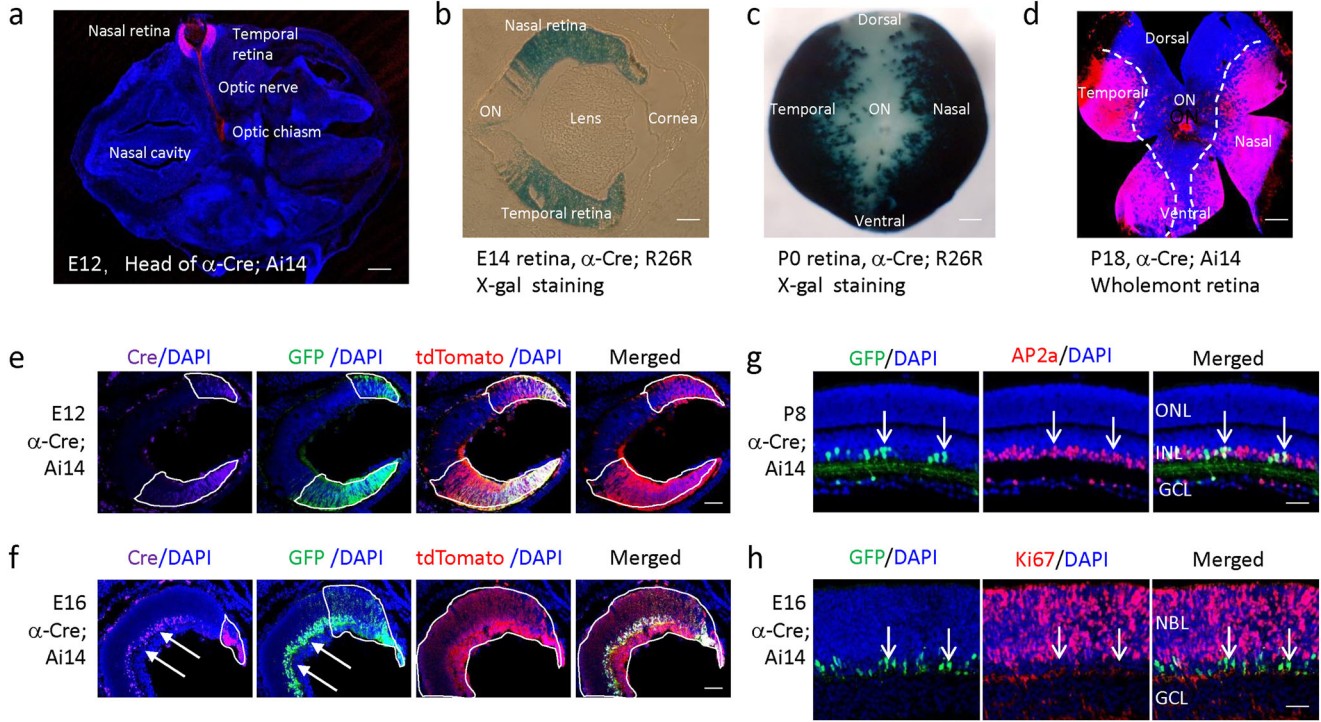

**Fig. 4 α-Cre activity in the retina. a** Horizontal sections of *α-Cre;Ai14* head at E12, showing tdTomato in the nasal and temporal peripheral retina as well as RGC axons in the optic nerve and optic chiasm. **b** X-gal staining of *α-Cre;R26R* eyeball at E14, showing α-Cre activity in the nasal and temporal peripheral retina. **c, d** X-gal staining of *α-Cre;R26R* whole retina at P0 (**c**), and tdTomato fluorescence in *α-Cre;Ai14* at P18 both showing α-Cre activity in the nasal and temporal peripheral retina but not the dorsal and ventral retina. **e** Horizontal sections of *α-Cre;Ai14* retina at E12, showing *α-Cre* expression (anti-Cre immunofluorescence), GFP expression (anti-GFP immunofluorescence), and lineage tracing (tdTomato fluorescence) in the nasal and temporal peripheral retina. **f** Horizontal sections of *α-Cre;Ai14* retina at E16, showing Cre and GFP in amacrine cells (white arrows). **g** AP2a staining (red) of P8 *α-Cre* retina, white arrows show typical GFP+; AP2a+ cells. **h** Ki67 staining (red) of E16 *α-Cre* retina; white arrows show typical GFP+;Ki67- cells. The white boxes in **e, f** represent the expression areas of Cre, GFP or tdTomato in retinal progenitors, respectively. ON, optic nerve; ONL, outer nucleus layer; INL, inner nucleus layer; GCL, ganglion cell layer; NBL, neuroblastic cell layer. Scale bars: **a** 200 μm; **c–h** 50 μm.

were tdTomato+, including the vomeronasal nerve projecting to accessory olfactory bulb (Fig. 5a, e).

We also examined the MOE as the *α-Cre* transgene was in the *Olfr291-290* and *Olfr300-292* clusters (Fig. 3i). Indeed, transgene activity was evident in many cells of the nasal olfactory epithelium (Fig. 5a). Many tdTomato+ cells clustered in the dorsal and lateral MOE (Fig. 5a). The MOE contains mature and immature OSNs, horizontal basal cells, globose basal cells, and sustentacular cells. Gα$_0$ is widely expressed in OSNs of the MOE[36,37]. Indeed, all tdTomato+ cells in the MOE were co-localized with Gα$_0$ (Fig. 5f). Olfactory marker protein (OMP) is a marker of mature OSNs[38]. Most tdTomato+ cells in the MOE were in areas with low expression of OMP at P2, but many of them were positive for OMP (Fig. 5g). Like the optic nerve (Fig. 4a) and the vomeronasal nerve (Fig. 5e), MOE OSN axons were tdTomato+, including the olfactory nerve projecting to olfactory bulb (Fig. 5h). We also detected GFP in VSN axons (Fig. 5e) and OSN axons (Fig. 5h), indicating that α-Cre is expressed in these mature neurons, akin to retinal amacrine cells (Fig. 4g). However, we did not detect tdTomato in the septal organ of Masera and the ganglion of Grüneberg, and we also had no evidence that α-Cre is expressed in OB or AOB neurons. Overall, these data reveal that α-Cre is active in some Gα$_0$+ basal VSNs and OSNs in the MOE.

**Expression of the *α-Cre* transgene does not match that of Pax6 in the olfactory system.** Our data are consistent with the idea that olfactory expression of α-Cre reflects the integration site. However, another possibility is that the Pax6 regulatory elements drive transgene expression in this tissue because Pax6 plays critical

roles in distinct developmental processes of the olfactory system. As the development of the olfactory placode is severely compromised, there are no olfactory receptor neurons in Pax6$^{-/-}$ mice (*Sey/Sey*)[39,40]. Pax6 is a master regulator of stem cell differentiation in the MOE[41,42], and is expressed throughout the development stages of the olfactory system[43,44]. In the MOE and VNO, Pax6 is expressed in non-neuronal linages as well as some OSN progenitors or precursors. For instance, in the MOE Pax6 is expressed in globose basal cell, horizontal basal cells, sustentacular cells and bowman's gland cell[41,45,46]. In the VNO, Pax6 is expressed in non-sensory epithelium, sustentacular cells and some VSN progenitors or precursors[46,47]. Thus, it is possible the regulatory elements for endogenous Pax6 gene expression in olfactory system may also drive the *α-Cre* transgene expression in MOE or VNO. Staining of Pax6 in the VNO/MOE of E16 and P2 α-Cre mice confirmed the reported Pax6 expression patterns (Fig. 6)[41,46,47]. In the VNO, Pax6 was mainly expressed in non-sensory epithelium (NSE), sustentacular cells and some basal VSNs, but was never co-localized with tdTomato (Fig. 6a, b). In the MOE, Pax6 was mainly expressed in sustentacular cells and some basal cells, but most of them were not co-localized with tdTomato (Fig. 6c, d). Occasionally very few Pax6+/tdTomato+ cells could be found (Fig. 6d). While we cannot formally exclude the possibility that activity of Pax6 regulatory elements earlier in development contributes to transgene expression, the absence of Pax6 in the vast majority of tdTomato+ neurons contrasts the similarity between the patterns of Cre activity and neighboring Olfr and Vmnr clusters, suggesting that the integration site plays a key role in driving α-Cre expression in the olfactory system.

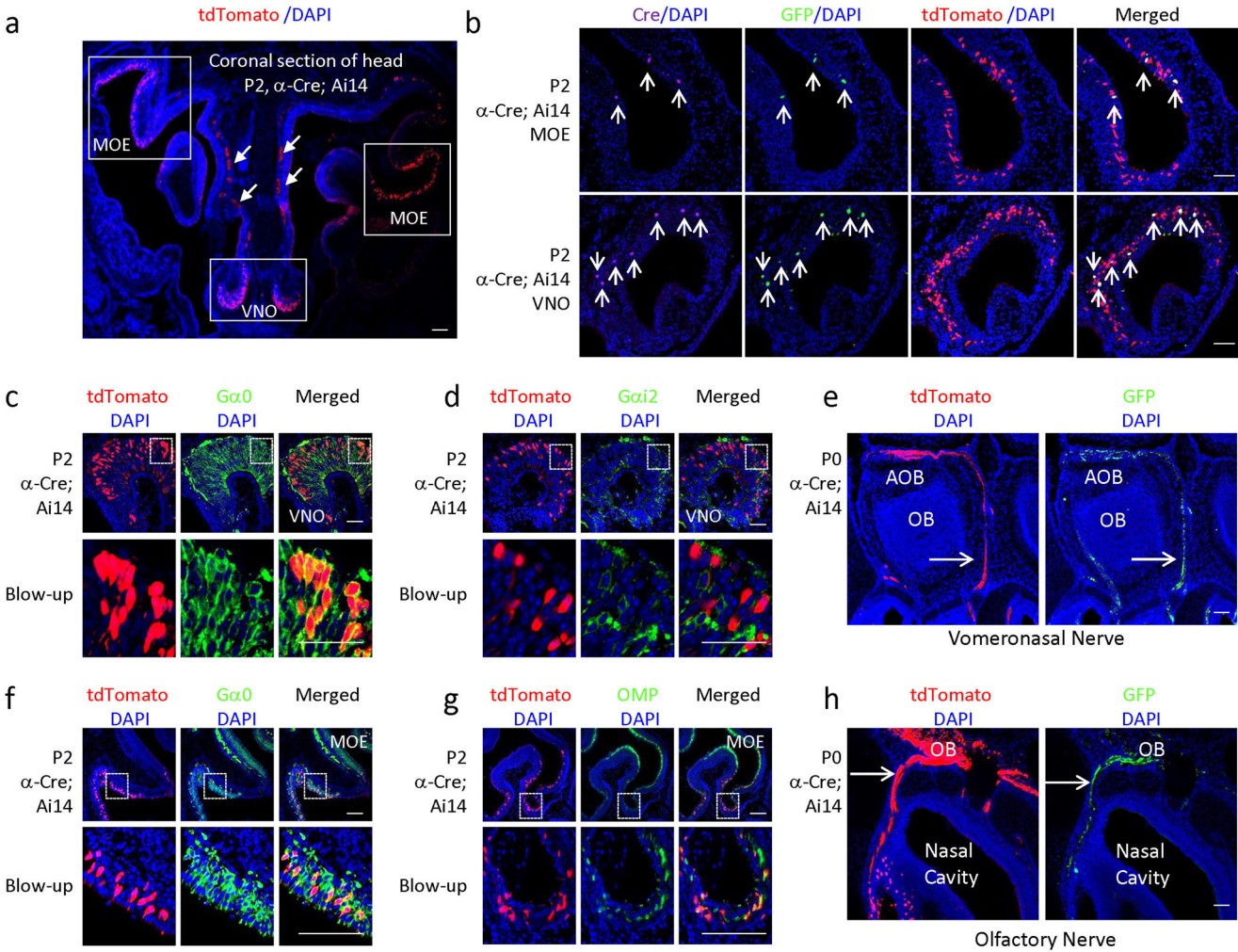

**Fig. 5 α-Cre activity in the olfactory system. a** Coronal section of the head of a P2 α-Cre;Ai14 mouse, showing tdTomato+ cells in VNO and MOE (highlighted in white boxes). White arrows indicate the vomeronasal nerve fibers. **b** Coronal MOE and VNO sections of P2 α-Cre;Ai14 mouse stained for Cre (purple), GFP (green), tdTomato (red) and nucleus (blue). White arrows indicate Cre+ cells colocalize with GFP + cells. **c**, **d** Coronal VNO sections of P2 α-Cre;Ai14 mouse stained for nucleus (blue), tdTomato (red), and Gα0 (green, **c**), or Gαi2 (green, **d**). Selected areas are blown up to show the colocalization between indicated proteins and tdTomato (dotted white box). **e** Horizontal sections of P0 α-Cre;Ai14 head showing α-Cre expression (anti-GFP) and lineage tracing (tdTomato) in vomeronasal nerve (white arrow) projecting to AOB. **f**, **g** Coronal head sections of P2 α-Cre;Ai14 mouse stained for nucleus (blue), tdTomato (red), and Gα0 (green, **f**), or OMP (green, **g**). Selected areas are blown up to show the colocalization between indicated proteins and tdTomato (dotted white box). **h** Horizontal sections of P0 α-Cre;Ai14 head showing α-Cre expression (anti-GFP) and lineage tracing (tdTomato) in olfactory epithelium, olfactory nerve (white arrow) projecting to olfactory bulb. GFP, green fluorescent protein; MOE, main olfactory epithelium; OMP, Olfactory marker protein; VNO, Vomeronasal organ; OB, olfactory bulb; AOB, accessory olfactory bulb. Scale bar: 200 μm in **a**, 100 μm in **e**/**h**, 40 μm in **c**, **d**, **f**, **g**.

**The α-Cre transgene has no significant effect on neighboring endogenous genes.** Expression of α-Cre in olfactory neurons reflects the regulatory elements of neighboring endogenous gene clusters on the transgene. Thus, we next assessed whether the corollary applies: does the α-Cre transgene affect the expression of olfactory genes? Using quantitative RT-PCR, we found that mRNA levels of 4 flanking *Vmn2r* genes (*Vmn2R67, 68, 69, 70*), 2 flanking *Olfr* genes (*Olfr291, Olfr300*) and the *OMP* (olfactory marker protein) were not different between α-Cre+ and α-Cre− mice in the retina and nasal mucosa (Fig. 7a). This result suggested that the α-Cre transgene has no significant effects on adjacent genes, although these data do not exclude more distal effects.

There are many chemosensory receptor genes on chromosome 7 (Fig. 3). Thus, to further explore the possibility of transgene interference, we employed the buried food pellet (BFP) test[48,49] to examine olfactory function. Based on the average mobility scores (quadrants/s), there was no significant difference between α-Cre+ and α-Cre− mice (p = 0.3567) (Fig. 7b). The average latency time to find buried pellets across three trials also showed no significant difference between α-Cre+ and α-Cre− mice (p = 0.1276) (Fig. 7c). Together, our molecular and behavioral studies suggest that the α-Cre transgene does not alter the expression of adjacent chemosensory receptor genes.

## Discussion

The α-Cre mouse is widely used in retinal research, e.g. refs. [3,5,6,50–53]. However, the integration site was unknown, precluding the development of genotyping primers from defining zygosity. It has also been unclear whether the transgene disrupts the expression of flanking genes. Previous work in our lab implied that α-Cre might lie on chromosome seven[7]. Indeed, here we mapped the insertion site of the α-Cre transgene to this chromosome using TLA[23], a valuable method to map integration

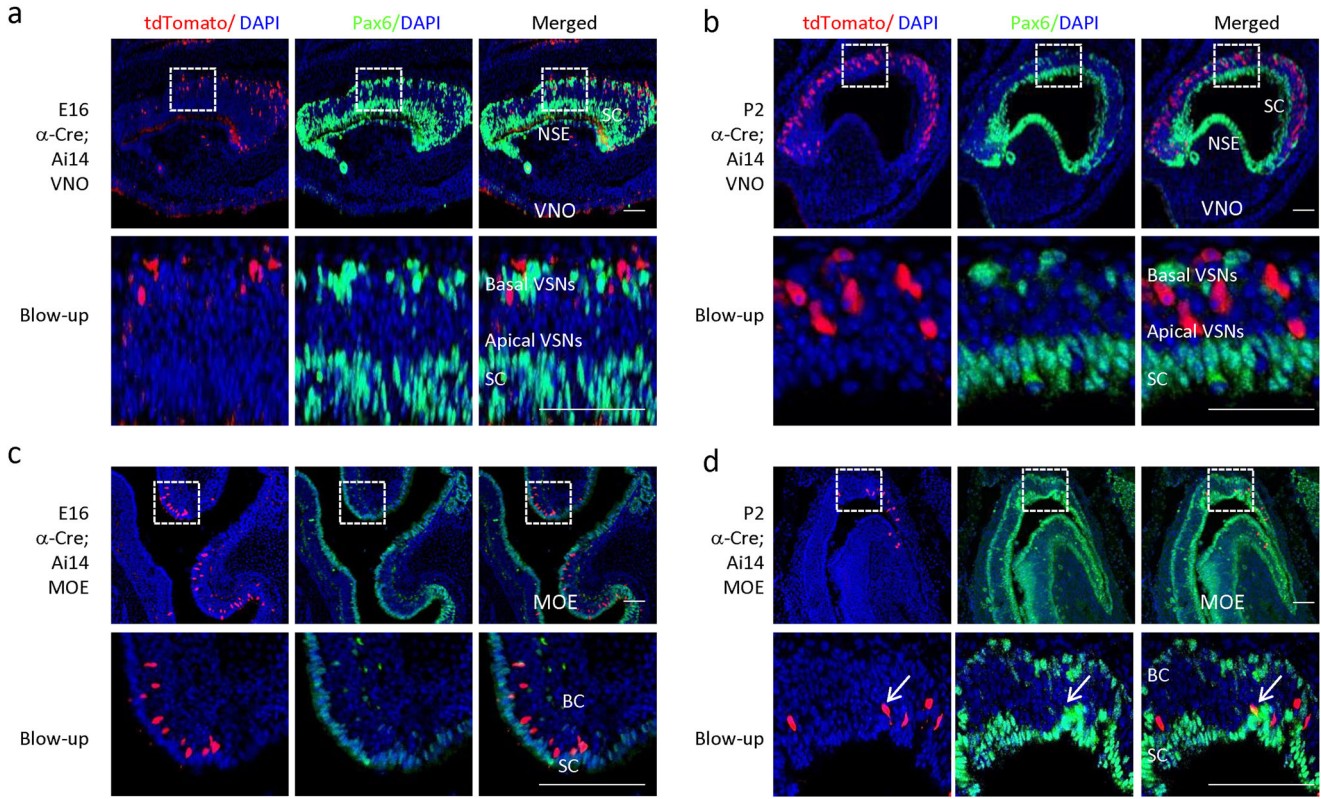

**Fig. 6 α-Cre does not colocalize with Pax6 in the olfactory system.** Coronal VNO (**a**, **b**) or MOE (**c**, **d**) sections of E16 or P2 *α-Cre;Ai14* mouse stained for Pax6 (green), tdTomato (red) and nucleus (blue). Selected areas are blown up to show the colocalization between Pax6 and tdTomato (dotted white box). White arrows in **d** indicate one Pax6+ and tdTomato+ cell. VNO, Vomeronasal organ; MOE, main olfactory sensory epithelium; NSE, non-sensory epithelium; SC, sustentacular cells; BC, Basal cells; VSNs, vomeronasal sensory neurons. Scale bar: 100 μm.

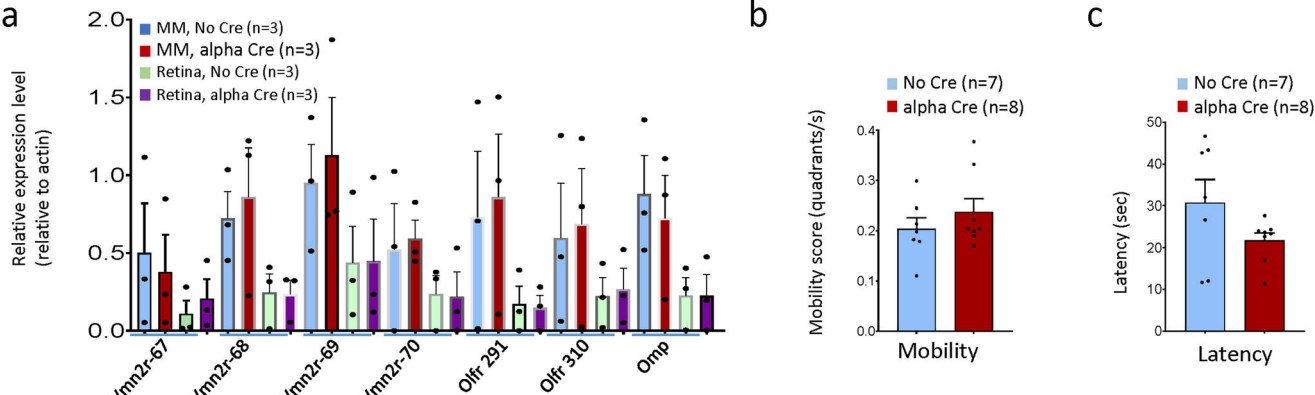

**Fig. 7 Effects of the *α-Cre* transgenes on olfactory gene expression and function. a** RT-PCR was used to assess expression of the indicated genes from nasal mucosa (NM) and retina of indicated genotypes at P13. **b**, **c** The mobility score (**b**) and the latency to find buried pellets (**c**) of the indicated genotypes at P30. Error bars represent SE of measurements and asterisks indicate significant differences between indicated genotypes (*$p < 0.05$, **$p < 0.01$, one-way ANOVA followed by Bonferroni correction).

sites[18–20]. The *α-Cre* transgene maps to 7qD3, in a cluster of *Vmn2r* GPCR genes, with loss of a 27 kb stretch. Relative to transcriptional direction, we found repetitive sequence upstream of the transgene, precluding the use of primers in this region to define zygosity. However, the downstream sequence was not repetitive, and we designed primer sets that successfully distinguished hemi- or homozygous animals. These tools will be helpful to labs working with *α-Cre* mice. Further study revealed there are four copies of *α-Cre* transgene in tandem. Where the 3′ end of the tandem copies meet chromosome 7 (the left end of the transgene), the *SV40 poly A* sequence is missing and terminates

with the *Hbb* intron. We are confident that all four copies are inserted at a single genomic site as TLA revealed only a single insertion site.

Transgene insertion can affect the expression of nearby endogenous genes[18]. In many cases, insertion mutates an endogenous gene, leading to severe defects or lethality[54]. However, there are no annotated genes in the *α-Cre* integration site. Nevertheless, transgenes can also integrate into endogenous regulatory elements or cause deletions and rearrangements that encompass such elements, and they may also indirectly alter chromatin structure[18,55–57]. Importantly, we did not detect

changes in the mRNA levels of seven olfactory-related genes flanking the α-Cre transgene, either in the nasal mucosa or retina. Moreover, a behavioral test did not detect any effect on olfactory function. Thus, we conclude that the α-Cre transgene has no significant effects on the expression of nearby endogenous genes.

The location of the transgene in a large cluster of olfactory genes led us to ask whether it is expressed in the olfactory system. The original α-Cre studies, based on GFP expression and Cre-dependent hPLAP/X-gal staining of embryos, did not report expression in non-retinal tissues[3,4]. Using two different Cre reporter systems, we confirmed that the temporal and spatial pattern of expression of α-Cre in our mice matched that of the prior studies[3,4]. We also found that α-Cre expression starts from E16 in post-mitotic AP2a[+] amacrine cells all over the retina (including the central retina) without any gradient. This expression is not silenced and is persistent in adult mice[5]. Thus, the expression of the α-Cre transgene in amacrine cells was much late than in retinal progenitors and may reflect the birthdate of these amacrine cells[58,59].

In addition to the classic retinal pattern, we found α-Cre activity in two murine olfactory subsystems, the MOE and VNO. The VNO exhibited more positive cells. Most α-Cre[+] cells are in the basal layer of the VNO, and all of them express Gα₀ but not Gαi2, confirming they are basal VSNs. These data are consistent with the proximity of the transgene to Vmn2R 68-69. The closest Olfr genes are Olfr290 and Olfr310, which are 0.3 Mb and 0.85 Mb from the α-Cre transgene, respectively. MOE olfactory sensory neurons (OSNs) generally only express one Olfr gene per cell[32], which may explain why only a few OSNs were positive for α-Cre activity. Many α-Cre[+] cells locate at the dorsal and lateral MOE, and all of them also express Gα₀, many of them are OMP[+], confirming they are OSNs. The vast majority of α-Cre[+] cells in the VNO and MOE do not express Pax6, consistent with the idea that α-Cre expression in olfactory neurons reflects the integration site.

In summary, our results provide detailed information on the integration site of the α-Cre transgene on chromosome 7, the internal sequences retained during integration, and expression in both the developing retina and olfactory system. Primers from the left integration junction are suitable for distinguishing heterozygotes from homozygotes. These results will aid future application of α-Cre mice in the retina and olfactory system research.

## Methods

**Mouse strains and genotyping.** Mice were treated according to institutional and national guidelines. All animal procedures were reviewed and approved by the Animal Care Committee (ACC) of University Health Network (UHN), Toronto, Ontario, Canada (AUP#1411), and the Ethical Review Committee of Animal Research of West China Hospital, Sichuan University, Chengdu, Sichuan province, China (AUP# 2018008A), and performed in compliance with the Association for Research in Vision and Ophthalmology (ARVO) statement for the use of animals in ophthalmic and visual research. α-Cre mice (P Gruss), R26R mice (the Jackson Laboratory, stock 003474), Ai14 mice (the Jackson Laboratory, stock 007914) were housed in the Animal Resources Centre of UHN (Toronto, Canada) and the Laboratory Animal Center of Sichuan University (Chengdu, China) with a normal 12-h/12-h light/dark schedule. Fresh water and rodent diet were always available. Age-matched wild type ICR mice were used as α-Cre negative (No Cre) control. Genotyping was performed as before[6,60] and as per The Jackson Laboratory guideline, and primers are listed in Supplementary Table 4. The α-Cre transgene zygosity was determined by progeny testing assays, following The Jackson Laboratory genotyping guidelines. Briefly, this involves mating α-Cre mice to wild-type mice and genotyping the offspring. If any of the offspring lack α-Cre, the parent is hemizygous Tg/+. If all the offspring test positive for α-Cre (at least 20 offspring were tested), the parent is homozygous Tg/Tg. Timed mating was set up according to the Jackson Laboratory's guide, and the day a vaginal plug was detected was considered as E0 of the pregnancy.

**Targeted locus amplification (TLA) and sequence alignment.** Spleen cells of α-Cre mice at P70 were isolated and prepared[23]. In brief, we isolate the spleen from a 10-week old mouse, put it in a 50 ml falcon tube with 10 ml 10%FCS/PBS. Disrupt

the spleen by pushing it gently through a 40 μm mesh that is placed on top of the 50 ml tube, rinse with 10%FCS/PBS to collect all cells. Centrifuge 250 x g at room temperature for 10 min. Discard the supernatant. Resuspend the pellet in 25 ml RBC lysis buffer and incubate 5 min at room temperature. Centrifuge 250 x g at room temperature for 10 min. Discard the supernatant. Resuspend the pellet in 5 ml 10%FCS/PBS. Count the cells and aliquot per 10 million cells in a 1.5 ml tube. Centrifuge 250 x g for 10 min. Carefully remove supernatant. Resuspend each pellet in 1 ml freezing buffer and transfer to a cryopreservation vial. Samples were stored at −80 °C until shipment to Cergentis (Utrecht, Netherlands) for TLA processing. TLA was performed according to Cergentis (Utrecht, Netherlands) standard protocol[23]. In brief, cells were crosslinked by formaldehyde and crosslinked genomic DNA is digested into ~0.2 kb small fragments by restriction enzyme NlaIII (recognition sequence CATG) and ligated to form larger circular DNA. These DNA fragments are de-crosslinked and digested by NspI (recognition sequence RCATGY) to create ~2 kb fragments, which are then subjected to PCR with primers derived from sequences unique to the Cre transgene (Supplementary Table 1). The PCR products were purified, and library prepped using the Illumina NexteraXT protocol and sequenced on an Illumina Miseq sequencer. Reads were mapped using Burrows-Wheeler Aligner's Smith-Waterman (BWA-SW). This allows partial mapping, which is optimally suited for identifying break spanning reads. Version Mm10 was used for mapping the mouse genome.

**Sequence validations.** Since breakpoint-spanning reads were available, two breakpoints and five linkers of the α-Cre transgene were confirmed by PCR amplification and Sanger sequencing. Primers were designed using Primer3 software (for primer sequences, see Supplementary Table 3).

Different primer combinations were tested by gel-electrophoresis. PCR products with optimal results were sent to Sangon Biotech (Shanghai, China) for Sanger sequencing.

**Genotyping of zygosity.** Based on the results of TLA and PCR validation, we developed several genotyping assays for the zygosity differentiation of α-Cre (Supplementary Table 2 and Fig. 2a). One assay included three primers: a shared forward genomic primer and two reverse primers designed specific to either the α-Cre or the wild-type allele.

**Copy number of transgenes.** Tail genomic DNA was isolated from 26 mice with homozygote α-Cre, 38 mice with heterozygote α-Cre and 13 mice without α-Cre (wild type control). Real-time quantitative PCR using qTOWER 2.2 PCR machine (Analytik Jena) or C1000 touch Thermal Cycler (Bio-Rad) was employed to measure the relative level of a 143 bp DNA fragment in Pax6 P0 promoter. Primers used are:

P0P F171: 5′-GGAGGACAATACCAGCCAGA-3′
P0P R312: 5′-GTTCAGCTCGGCAGATTAGC-3′

Tests were run in duplicate with EvaGreen PCR Supermix (SsoFast, Bio-Rad) or SYBR Green PCR Master Mix (Applied Biosystems, Thermo Fisher Scientific). PCR consisted of 40 cycles of denaturation at 95 °C for 15 s and annealing and extension at 55 °C for 30 s. An additional cycle (95 °C, 15 s) generated a dissociation curve to confirm a single product. Target gene relative expression was quantified according to the comparative ΔCT method. The signal levels were normalized against the housekeeping gene β-actin, and the average copy number of the Pax6 P0 promoter of wild type mice was set as 2. Then the copy number of the Pax6 P0 promoter of α-Cre mice was calculated based on the relative signal levels. The copy number of α-Cre transgenes was the average copy number of Pax6 P0 promoter minus 2.

**Immunofluorescence and X-gal staining.** Eyeballs or heads were fixed for 1 h at 4 °C in 4% paraformaldehyde, embedded in OCT (TissueTek 4583), frozen on dry ice and cut into 12–14 μm sections on Superfrost slides. The following antibodies were used: Ap2a (1:500, Santa Cruz, SC-8975), Cre (1:500, Cell signaling, 15036), GFP (1:500, Abcam, ab6673), Gα0 (1:500, Upstate, 07–634), Gαi2 (1:500, Abcam, ab157204), Ki67 (1:1000, BD science Pharmingen, 14-5698-82), mCherry (1:1000, Novus Biological, NBP2-25158), OMP (1:500, Abcam, ab183947), Pax6 (1:500, Covance, PRB-278P). Antigen retrieval was performed as described[60]. Primary antibodies or labeled cells were visualized using donkey anti-rat, donkey anti-chicken, donkey anti-goat and donkey anti-rabbit antibodies conjugated with Alexa-488, Alexa-568 or Alexa-647 (1:1000; Molecular Probes). Nuclei were counter-stained with DAPI (1:200, 1 mg/ml, Sigma, D9542) and mounted with Mowiol. Validations of the primary antibodies are provided on the manufacturers' websites or in the referenced citations.

For whole-mount α-Cre, Ai14 mouse retinal staining, eyeballs were enucleated and incubated for 30 min in 4% paraformaldehyde. The retinas were incubated DAPI in PBS for 30 min. X-gal (β-galactosidase) staining protocol is from Soriano Lab[29]. Retinal sections and whole-mount retinas of α-Cre, R26R mice were incubated with X-gal staining solution (1 mg/ml X-Gal, 5 mM potassium ferricyanide, 5 mM potassium ferrocyanide, 2 mM MgCl2, 0.01% sodium deoxycholate, and 0.02% Nonidet P-40 in PBS) in dark for 2 h at 37 °C. After briefly washes with PBS, radial cuts were made to divide the retina into four quadrants to flatten the retina, and flat retinas and retinal section slides were mounted with Mowiol. Stained sections and slides were analyzed using a Zeiss Axio

Imager Z2 fluorescence microscope and Zeiss Zen software. Color images were taken using an Olympus BX61 microscope.

**RNA extraction, reverse transcription, and quantitative real-time PCR.** Total RNA was isolated from P13 nasal mucosa and retinas using TRIzol reagent (Invitrogen, Thermo Fisher Scientific) followed by digestion with RNase-free DNase (DNA-free, Invitrogen, Thermo Fisher Scientific) to remove DNA contamination. Integrity and quantity of the RNA was evaluated by a NanoDrop (NanoDrop Technologies). Reverse transcription PCR assays were performed using a RT Reagent Kit with gDNA Eraser (TaKaRa). Real-time quantitative PCR using qTOWER 2.2 PCR machine (Analytik Jena) was employed to measure the expression level of seven genes. Primers used are listed in Supplementary Table 5. Tests were run in duplicate on 3 separate biological samples with SYBR Green PCR Master Mix (Applied Biosystems, Thermo Fisher Scientific). PCR consisted of 40 cycles of denaturation at 95 °C for 15 s and annealing and extension at 55 °C for 30 s. An additional cycle (95 °C, 15 s) generated a dissociation curve to confirm a single product. Target gene relative expression was quantified according to the comparative ΔCT method. Transcription levels were normalized against the housekeeping gene β-actin.

**Olfactory function and mobility measurements.** Mouse olfactory function and mobility were measured by the buried food pellet (BFP) test as previously described[48,49,61]. Animals were tested at ~4 weeks of age. The mice were kept in individual cage (home cage) and food deprived overnight with access to water ad libitum. Mice were habituated for 30 min by placing their home cage in the testing room. Then mice were introduced into a clean habituation cage (470 × 312 × 260 mm) for 10 min. This habituation step was repeated for three times with a new habituation cage each time. For the third time, the habituation cage was filled with 3 cm-thick fresh corn-cob bedding. Then, the mouse was transferred back to its home cage for 10 min and ready for test.

The test cage (size 470 × 312 × 260 mm) was filled with 3-cm thick fresh bedding. A standard lab diet pellet (weight 1 ± 0.1 g) was buried ~0.5 cm under the surface of the bedding. Set a timer for 5 min. The mouse to be tested was re-introduced into the test cage with head facing the cage wall. Time from the introduction of the animal in the cage until it retrieved the food pellet with its front paws was measured in seconds up to a maximum of 300 s (5 min). After each test, bedding was changed, and test cage were cleaned with animal room cleaning solution. One mouse was tested for three times with 10-min intervals in its home cage. The sites of animal placement were always the same, but the location of the food pellet was changed for each test in a random fashion.

To assess the mobility of mice, the test cage was divided into four equal quadrants and each movement of the mouse from one quadrant into a neighboring quadrant was scored, whenever more than half of its body length (excluding the tail) had entered the new quadrant. Mobility levels were scored as quadrants/s.

**Statistics and reproducibility.** All data are expressed as the means ± SE. For α-Cre transgene copy number analysis, 13 wild type, 38 heterozygote, and 26 homozygote mice were used. For RT-PCR analysis of flanking genes, 3 wild type and 3 α-Cre transgene mice were used. For olfactory function and mobility measurements, 7 wild type and 8 α-Cre transgene mice were used. Statistical analyses were performed using GraphPad Prism version 9.0 (GraphPad Software). The differences between mean values were evaluated using a one-way ANOVA test. A $p$-value less than 0.05 were considered to indicate a statistically significant difference between means.

**Reporting summary.** Further information on research design is available in the Nature Research Reporting Summary linked to this article.

## Data availability

The α-Cre transgene sequence and flanking chromosome 7 sequence have deposited to GenBank, accession number: MZ890862. Uncropped gel images for Figs. 2b, d, 3c–e and Supplementary Fig 1a are provided as Supplementary Fig 1b, 2, and 3. The source data behind the Figs. 3g, 7a–c in the paper are in Supplementary Data 1. All other data are available from the corresponding author on reasonable request.

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

## Acknowledgements

This work was supported by the National Natural Science Foundation of China (81870665, 82171063) to DC, and grants from the Canadian Institutes for Health Research (153128, 173474) to RB. The funders had no role in study design, data collection and analysis, decision to publish, or preparation of the manuscript.

## Author contributions

Conception and design: D.C. and R.B. Analysis and interpretation: Y.F., W.C., R.W., D.C., R.B. Data collection: Y.F., W.C., R.W., W.Q., J.P., T.Y., D.C. Obtain funding: D.C. and R.B. Overall responsibility: D.C. and R.B. Manuscript writing and editing: Y.F., W.C., R.W., D.C., R.B. All authors read and approved the final manuscript.

## Competing interests

The authors declare no competing interests.
