## [Peer Review File · Communications Biology]

Reviewers' comments:

Reviewer #1 (Remarks to the Author):

A transgenic mouse line called Pax-a-cre has been used in many studies of retinal development. In this line, Cre recombinase and GFP are expressed under the control of nominally retina-specific regulatory elements from the Pax6 gene. Here, Fan et al. use a method called targeted locus amplification (TLA) to map the insertion site of this transgene. They show that four copies were integrated at a single site within a large group of olfactory and vomeronasal receptor (Figures 1-3). Presumably for this reason, retinal expression is accompanied by expression in olfactory and vomeronasal sensory neurons, as demonstrated by use of a floxed reporter gene (Figure 5). The authors also confirm previously reported expression in retina (Figure 4). On the other hand, they find no evidence that the transgene affects expression of neighboring olfactory or vomeronasal receptor genes (Figure 6).

These studies are performed to a high technical standard and presented clearly. My two comments are on somewhat tangential issues that the editor should be free to consider or ignore.

First, the primary value of the results presented here is to enable generation of genotyping probes that would allow distinction between homozygotes and heterozygotes. As the authors state: "The α -Cre mouse is widely used in retinal research. However, the integration site was unknown, precluding the development of genotyping primers from defining zygosity." That will be of interest to a very small audience. Compounding this problem, there is no demonstration of this genotyping ability, which might be a requirement given the rationale. (They do estimate copy number, but that could have been done without knowing the insertion site.)

Second, the paper is clearly inspired by a previous one (ref. 18) but that debt is poorly acknowledged. In that paper, Laboulaye et al., begin with the observation that they cannot generate mice that contain both a transgene (HB9-GFP) and a deletion allele (CDH6), suggesting linkage. Here, the authors begin with the observation that they cannot generate mice that contain both a transgene (α Cre) and a deletion allele (BAX), suggesting linkage. Laboulaye go on to note that methods for identifying transgene insertion sites in mice have been unreliable ("available methods ... have been cumbersome, little used and, in our hands, largely unsuccessful"). The current authors make the same point: "...these methods are cumbersome and largely unsuccessful...." Both papers then state that TLA provides a better alternative, and explain it. Laboulaye: "In TLA, genomic DNA in nuclei is cross-linked by formaldehyde, digested into small fragments by the frequently cutting NlaIII restriction enzyme and religated to form larger circular DNA containing fragments that were likely to have been near neighbors on a chromosome. These fragments are de-crosslinked and digested by another restriction enzyme, NspI, to create ~2kb fragments, which are then subjected to PCR with primers derived from sequences unique to the transgene." Present paper: "In TLA, crosslinked genomic DNA is digested into ~0.2 kb small fragments by the restriction enzyme NlaIII and ligated to form larger circular DNA. These DNA fragments are de-crosslinked and digested by NspI to create ~2 kb fragments, which are then subjected to PCR with primers derived from sequences unique to the transgene." Later, Laboulaye state that they "next asked if the transgenes we had studied affected expression of neighboring endogenous genes." Likewise, the present paper "next assessed whether the corollary applies: does the α -Cre transgene affect the expression of olfactory genes?" And so on. I do NOT think this rises to the level of plagiarism, and I am very definitely NOT suggesting any misconduct. On the other hand, it is not a very generous way to acknowledge a debt to predecessors.

Reviewer #2 (Remarks to the Author):

In "Mapping Transgene Insertion Sites Reveals the alpha-Cre Transgene Expression in Both Developing Retina and Olfactory Neurons," Fan and colleagues further characterize the genomic location and expression pattern of the widely used alpha-Cre transgenic mouse line. The genomic insertion site of the alpha-Cre transgene was mapped to Chromosome 7 using targeted locus amplification. Further analysis precisely identified the left and right junction of insertion as well the copy number (4) and array of each copy. Due to a 500bp repeat sequence located at the right junction, only the left junction can be used for determining hemi or homozygosity. The authors then verified the alpha-cre line in their lab expresses within the retina as was previously identified. Upon further analysis the authors discover expression of the transgene outside of the retina in the developing olfactory system and suggest this expression is due to position effect of its location on chromosome 7. Lastly, the authors verify the transgene does not influence the expression of the surrounding olfactory genes using quantitative RT-PCR or the ability of the mice to find food using the buried food pellet test.

Due to the extensive use of the alpha-Cre transgene, the science community will benefit from the further characterization presented by Fan et al. This manuscript is likely to be cited along with the original publication of the transgene, and potentially the last in depth look at its genomic location and expression pattern. Despite its potential the authors need to further explore the olfactory system expression of the alpha-cre transgene. Therefore, the authors should perform some additional experiments to fully validate their conclusions under a major revision.

Major Comments

1. There is an abundance of literature describing the importance and expression pattern of Pax6 during olfactory development and into adulthood. None of this literature is cited or discussed within the manuscript. The authors should familiarize themselves with this literature and include its key findings.
2. Related to point 1, because Pax6 is essential and expressed in the nasal tissues it is possible the transgene expression identified is due to endogenous Pax6 expression, even though the alpha regulatory region is supposed to be retina specific. The authors should perform immunohistochemistry staining for Pax6 and assess for co-localization with Cre, GFP, and tdTomato. Cre should be included as a control because of the architecture of the transgene. The use of an IRES does not guarantee 1:1 expression between the first and second genes (reference below). With GFP being the second gene, Cre represents a more accurate readout.

Hiroyuki Mizuguchi, Zhili Xu, Akiko Ishii-Watabe, Eriko Uchida, Takao Hayakawa, IRES-Dependent Second Gene Expression Is Significantly Lower Than Cap-Dependent First Gene Expression in a Bicistronic Vector, *Molecular Therapy*, 2000

3. The authors rely on anatomical markers to evaluate transgene expression within the Vomeronasal organ, olfactory bulb, and accessory olfactory bulb in Figure 5. From these images it is unclear the cell types which contain transgene expression. The authors should show co-localization between the markers for the transgene (Cre, tdTomato) and known markers for the different cell types of these regions using either mRNA or protein co-localization techniques. For example, the VNO contains multiple different cell types and it is important to identify where the transgene is expressed (example references below). Specifically, with the suggestion the closest *Vmnr2* genes to the transgene are influencing expression, these should be included in the analysis. These experiments would benefit from coronal rather than horizontal sections as is the standard in the field for visualizing the VNO.

Isabelle Chang, Marta Parrilla, Expression patterns of homeobox genes in the mouse vomeronasal organ at postnatal stages, *Gene Expression Patterns*, 2016

Katreddi RR, Forni PE. Mechanisms underlying pre- and postnatal development of the vomeronasal organ. *Cell Mol Life Sci.* 2021;78(12):5069-5082. doi:10.1007/s00018-021-03829-3

Minor Comments

1. Why is there noise all along chromosome 7 and no other chromosome in Figure 1 b? It would be helpful to show the same image for the second TLA primer set used either in the main figure or supplemental.

Detailed point-by-point responses to Reviewer 1 and 2

Reviewer #1

1-A transgenic mouse line called Pax-a-cre has been used in many studies of retinal development. In this line, Cre recombinase and GFP are expressed under the control of nominally retina-specific regulatory elements from the Pax6 gene. Here, Fan et al. use a method called targeted locus amplification (TLA) to map the insertion site of this transgene. They show that four copies were integrated at a single site within a large group of olfactory and vomeronasal receptor (Figures 1-3). Presumably for this reason, retinal expression is accompanied by expression in olfactory and vomeronasal sensory neurons, as demonstrated by use of a floxed reporter gene (Figure 5). The authors also confirm previously reported expression in retina (Figure 4). On the other hand, they find no evidence that the transgene affects expression of neighboring olfactory or vomeronasal receptor genes (Figure 6). These studies are performed to a high technical standard and presented clearly.

Response: Thank you for this precise summary of our study, and very positive comments.

2-My two comments are on somewhat tangential issues that the editor should be free to consider or ignore.

Response: Thank you for these comments, which we address below.

3-First, the primary value of the results presented here is to enable generation of genotyping probes that would allow distinction between homozygotes and heterozygotes. As the authors state: "The α -Cre mouse is widely used in retinal research. However, the integration site was unknown, precluding the development of genotyping primers from defining zygosity." That will be of interest to a very small audience. Compounding this problem, there is no demonstration of this genotyping ability, which might be a requirement given the rationale. (They do estimate copy number, but that could have been done without knowing the insertion site.)

Response: We apologize for not presenting this clearly enough. In the original version, we presented primers in Figure 2 and Table 1. Now we present this data in an independent section "Genotyping of zygosity of the α -Cre transgene" (page 6, line 142-151) and an additional Figure (Figure 2F).

4-Second, the paper is clearly inspired by a previous one (ref. 18) but that debt is poorly acknowledged. In that paper, Laboulaye et al., begin with the observation that they cannot generate mice that contain both a transgene (HB9-GFP) and a deletion allele (CDH6), suggesting linkage. Here, the authors begin with the observation that they cannot generate mice that contain both a transgene (α Cre) and a deletion allele (BAX), suggesting linkage. Laboulaye go on to note that methods for identifying transgene insertion sites in mice have been unreliable (“available methods ... have been cumbersome, little used and, in our hands, largely unsuccessful”). The current authors make the same point: “...these methods are cumbersome and largely unsuccessful...” Both papers then state that TLA provides a better alternative, and explain it. Laboulaye: “In TLA, genomic DNA in nuclei is cross-linked by formaldehyde, digested into small fragments by the frequently cutting NlaIII restriction enzyme and religated to form larger circular DNA containing fragments that were likely to have been near neighbors on a chromosome. These fragments are de-crosslinked and digested by another restriction enzyme, NspI, to create ~2kb fragments, which are then subjected to PCR with primers derived from sequences unique to the transgene.” Present paper: “In TLA, crosslinked genomic DNA is digested into ~0.2 kb small fragments by the restriction enzyme NlaIII and ligated to form larger circular DNA. These DNA fragments are de-crosslinked and digested by NspI to create ~2 kb fragments, which are then subjected to PCR with primers derived from sequences unique to the transgene.” Later, Laboulaye state that they “next asked if the transgenes we had studied affected expression of neighboring endogenous genes.” Likewise, the present paper “next assessed whether the corollary applies: does the α -Cre transgene affect the expression of olfactory genes?” And so on. I do NOT think this rises to the level of plagiarism, and I am very definitely NOT suggesting any misconduct. On the other hand, it is not a very generous way to acknowledge a debt to predecessors.

Response: The reviewer is right; we were inspired the work done by Dr. Josh Sanes and colleagues (Ref 18) and Dr. Cox and Warming (Ref 19). This was acknowledged as “Inspired by these works, especially the work identified the insertion sites for three transgenes which label some retinal cells¹⁸ and for seven Cre transgenes¹⁹, we decided to use TLA to search the insertion site of the α -Cre transgene.” (Page 4, line 85-87). The descriptions of the details of the TLA technology were deleted from the introduction now.

Reviewer #2

1-In "Mapping Transgene Insertion Sites Reveals the alpha-Cre Transgene Expression in Both Developing Retina and Olfactory Neurons," Fan and colleagues further characterize the genomic location and expression pattern of the widely used alpha-Cre transgenic mouse line. The genomic insertion site of the alpha-Cre transgene was mapped to Chromosome 7 using targeted locus amplification. Further analysis precisely identified the left and right junction of insertion as well the copy number (4) and array of each copy. Due to a 500bp repeat sequence located at the right junction, only the left junction can be used for determining hemi or homozygosity. The authors then verified the alpha-cre line in their lab expresses within the retina as was previously identified. Upon further analysis the authors discover expression of the transgene outside of the retina in the developing olfactory system and suggest this expression is due to position effect of its location on chromosome 7. Lastly, the authors verify the transgene does not influence the expression of the surrounding olfactory genes using quantitative RT-PCR or the ability of the mice to find food using the buried food pellet test.

Response: Thank you for this precise summary of our study.

2-Due to the extensive use of the alpha-Cre transgene, the science community will benefit from the further characterization presented by Fan et al. This manuscript is likely to be cited along with the original publication of the transgene, and potentially the last in depth look at its genomic location and expression pattern.

Response: Thank you for these very positive comments.

3-Despite its potential the authors need to further explore the olfactory system expression of the alpha-cre transgene. Therefore, the authors should perform some additional experiments to fully validate their conclusions under a major revision.

Response: We thank the reviewer for their encouraging comments and for their careful critique. We have addressed their comments in depth below.

4- Major Comments-1: There is an abundance of literature describing the importance and expression pattern of Pax6 during olfactory development and into adulthood. None of this literature is cited or discussed within the manuscript. The authors should familiarize themselves with this literature and include its key findings.

Response: Thank you for this suggestion. The function and expression pattern of Pax6 in the major olfactory sensory epithelium (MOE)/ the vomeronasal organ (VNO) are described in a new section “Expression of the α -Cre transgene does not match that of Pax6 in the olfactory system” (Page 12, line 296-322), and 9 major papers are cited (ref 39-47).

5- Major Comments-2: Related to point 1, because Pax6 is essential and expressed in the nasal tissues it is possible the transgene expression identified is due to endogenous Pax6 expression, even though the alpha regulatory region is supposed to be retina specific. The authors should perform immunohistochemistry staining for Pax6 and assess for co-localization with Cre, GFP, and tdTomato. Cre should be included as a control because of the architecture of the transgene. The use of an IRES does not guarantee 1:1 expression between the first and second genes (reference below). With GFP being the second gene, Cre represents a more accurate readout.

Hiroyuki Mizuguchi, Zhili Xu, Akiko Ishii-Watabe, Eriko Uchida, Takao Hayakawa, IRES-Dependent Second Gene Expression Is Significantly Lower Than Cap-Dependent First Gene Expression in a Bicistronic Vector. *Molecular Therapy*, 2000

Response: We thank the reviewer for this helpful suggestion.

- 1) We used antibodies for Cre (Cell signaling, 15036, rabbit), GFP (Abcam, ab6673, goat), mCherry (Novus Biological, NBP2-25158, chicken), and Pax6 (Covance, PRB-278P, rabbit) to address this issue.
- 2) By co-staining of Cre, GFP and mCherry, we found that there are always more tdTomato+ cells than Cre+ or GFP+ cells in E12/E16 α -Cre retinas (Figure 4E-F), as well as P2 MOE/VNO (Figure 5B). The text has been modified to reflect the additional data (Page 9, line 218-233; Page 10, line 259-269).
- 3) Regarding Cre and GFP, in E12 retina they were almost identical (Figure 4E), but in E16 retina, GFP+ cells were more than Cre+ cells (Figure 4F), suggesting that the levels of Cre may be lower than GFP

in these cells (e.g. due to protein half-life differences and/or differences in sensitivity of detection). In the olfactory system (MOE/VNO), the GFP+ cells are all also Cre+, but both were much less than tdTomato+ cells (Figure 5B). The text has been modified to describe all these new data (Page 9, line 218-233; Page 10, line 259-269).

- 4) We also co-stained Pax6 (rabbit antibody) and tdTomato (chicken antibody for mCherry) to address if Pax6 colocalize with α -Cre. As shown in Figure 6, in both MOE/VNO at embryonic stage (E16) and postnatal stage (P2), most tdTomato⁺ cell were not co-localized with Pax6. We modified the text to describe these new data, including the cautious conclusion that it remains possible that prior activity of Pax6 elements earlier in development may contribute to some olfactory expression (page 12, line 296-322). We thank the reviewer for highlighting this important caveat.

6- Major Comments-3: The authors rely on anatomical markers to evaluate transgene expression within the Vomeronasal organ, olfactory bulb, and accessory olfactory bulb in Figure 5. From these images it is unclear the cell types which contain transgene expression. The authors should show co-localization between the markers for the transgene (Cre, tdTomato) and known markers for the different cell types of these regions using either mRNA or protein co-localization techniques. For example, the VNO contains multiple different cell types and it is important to identify where the transgene is expressed (example references below). Specifically, with the suggestion the closest Vmnr2 genes to the transgene are influencing expression, these should be included in the analysis. These experiments would benefit from coronal rather than horizontal sections as is the standard in the field for visualizing the VNO.

Isabelle Chang, Marta Parrilla, Expression patterns of homeobox genes in the mouse vomeronasal organ at postnatal stages, Gene Expression Patterns, 2016

Katreddi RR, Forni PE. Mechanisms underlying pre- and postnatal development of the vomeronasal organ. Cell Mol Life Sci. 2021;78(12):5069-5082. doi:10.1007/s00018-021-03829-3

Response: We thank the reviewer for drawing our attention to these helpful studies and cell type markers.

- 1) We used coronal sections of α -Cre mouse head to address this issue (Figure 5A-D, F-G).

- 2) The following antibodies were used: $G\alpha_0$ (Upstate, 07-634, rabbit), $G\alpha_{i2}$ (Abcam, ab157204, rabbit), mCherry (Novus Biological, NBP2-25158), OMP (Abcam, ab183947, rabbit).
- 3) In VNO, tdTomato⁺ cells locate at the basal layer (Figure 5A-D), and all express $G\alpha_0$ (Figure 5C) but not $G\alpha_{i2}$ (Figure 5D), indicating they are basal vomeronasal sensory neurons.
- 4) In MOE, tdTomato⁺ cells locate just beneath the apical sustentacular cell layer (Figure 6C-D), and all express $G\alpha_0$ (Figure 5F). Many of them are also OMP⁺, which is the mature marker of OSNs (Figure 5G).
- 5) We had no evidence that α -Cre is expressed in OB or AOB neurons.

The text has been modified to explain all the additional data using these new markers (Page 11, line 271-288).

7 Minor Comments-1: Why is there noise all along chromosome 7 and no other chromosome in Figure 1 b? It would be helpful to show the same image for the second TLA primer set used either in the main figure or supplemental.

Response:

- 1) The Cergentis Company did not provide us the whole genome coverage plot for the second TLA primer set, and just stated in the report that “similar results were obtained with primer set 2”.
- 2) TLA selectively amplifies and sequences entire genes based on the crosslinking of physically proximal sequences, thus the noise along chromosome 7 reflects intra-chromosomal contacts. Actually, there are also some minor signals along chromosome 1, 2, 4, 9 and 15, which may reflect the inter-chromosomal contacts. Both these phenomena are widely described in studies focused on chromosomal topology.

REVIEWERS' COMMENTS:

Reviewer #1 (Remarks to the Author):

The revisions have addressed all of my concerns

Reviewer #3 (Remarks to the Author):

In the resubmission by Fan et. al., the authors adequately addressed the reviewers concerns with regards to additional controls, genotyping, and providing credit to previous literature. The included data analyzing GFP, Cre, tdTomato and Pax6 expression in the developing retina and/or MOE/VNO provides greater context to the dynamics and lineage of the alpha-Cre transgene, a vital addition to readers. These additions provide stronger support for the environment surrounding the transgene insertion site influencing the expression pattern of alpha-Cre in the MOE/VNO. In addition, the more refined analysis of the MOE/VNO with specific markers helps clarify the described expression pattern.

My recommendation is to publish "Mapping Transgene Insertion Sites Reveals the α -Cre Transgene Expression in Both Developing Retina and Olfactory Neurons."